# Experimental Research on Deformation Characteristics of Using Silty Clay Modified Oil Shale Ash and Fly Ash as the Subgrade Material after Freeze-Thaw Cycles

**Haibin Wei** [1], **Qinglin Li** [1], **Leilei Han** [1], **Shuanye Han** [1], **Fuyu Wang** [1,*], **Yangpeng Zhang** [2,*] **and Zhao Chen** [3]

[1] School of Transportation, Jilin University, Changchun 130022, China; weihb@jlu.edu.cn (H.W.); liql1150142@163.com (Q.L.); hanll18@mails.jlu.edu.cn (L.H.); hansy18@mails.jlu.edu.cn (S.H.)

[2] Guangxi Transportation Research & Consulting Co., LTD; Guangxi Key Lab of Road structure and materials; High-grade highway construction & maintenance technology, materials and equipment transportation industry R&D center (Nanning), Nanning 530000, China

[3] Jilin Province Highway Group Co., LTD., Changchun 130025, China; chenzhao_2000@163.com

* Correspondence: wfy@jlu.edu.cn (F.W.); yangpengz16@mails.jlu.edu.cn (Y.Z.); Tel: +86-0431-8509-5370 (F.W.)

**Abstract:** To achieve the purposes of disposing industry solid wastes and enhancing the sustainability of subgrade life-cycle service performance in seasonally frozen regions compared to previous research of modified silty clay (MSC) composed of oil shale ash (OSA), fly ash (FA), and silty clay (SC), we identified for the first time the axial deformation characteristics of MSC with different levels of cycle load number, dynamic stress ratio, confining pressure, loading frequency, and F-T cycles; and corresponding to the above conditions, the normalized and logarithmic models on the plastic cumulative strain prediction of MSC are established. For the effect of cycle load number, results show that the cumulative plastic strain of MSC after 1, 10, and 100 cycle loads occupies for 28.72%~35.31%, 49.86%~55.59%, and 70.87%~78.39% of those after 8000 cycle loads, indicating that MSC possesses remarkable plastic stability after 100 cycles of cycle loads. For the effect of dynamic stress ratio, confining pressure, loading frequency, and F-T cycles, results show that dynamic stress ratio and F-T cycles are important factors affecting the axial deformation of MSC after repeated cycle loads; and under the low dynamic stress ratio, increasing confining pressure and loading frequency have insignificant effect on the axial strain of MSC after 8000 loads. In term of the normalized and logarithmic models on the plastic cumulative strain prediction of MSC, they have a high correlation coefficient with testing data, and according to the above models, the predicted result shows that the cumulative plastic strain of MSC ranges from 0.38 cm to 2.71 cm, and these predicted values are within the requirements in the related standards of highway subgrades and railway, indicating that the cumulative plastic strain of MSC is small and MSC is suitable to be used as the subgrade materials.

**Keywords:** modified silty clay; oil shale ash; fly ash; cumulative plastic strain; normalized and logarithmic models; freeze-thaw cycles

## 1. Introduction

Oil Shale is an important energy resource. Shale oils can be obtained by carbonization, which are ideal substitutes for conventional petroleum and gas. The global explored oil shale reserves exceed 10 trillion tons, and calculating at 5% shale oil content, the world has 500 billion tons of shale oil, nearly twice as much as oil reserves (about 270 billion tons) [1,2]. Even with lower oil rates, there are hundreds of billion tons of converted shale oil. Hence, many countries have included oil shales in their national energy reserve systems. The utilization of oil shales has a history of 200 years. However, the utilization of oil shale is still trapped by high costs and large quantities of byproduct wastes. Estonia has the most advanced oil shale utilization technology [3]. Its oil shale solid wastes exceed $8 \times 10^9$ tons, which seriously pollute the forests, lakes, and land environment in the northeastern regions [4,5]. The same situation occurs in China. There are $214 \times 10^8$ tons of oil shale solid wastes in Maoming oil shale mining area of Guangdong, China during the 70 years of production [6].

Significant progress has been made to utilize the OSA (oil shale ash) in economic ways. In view of positive effect of industrial residues in material improvement, adding OSA to construction or building materials has been concentrated by many researchers in recent years. Findings verify that concrete, bituminous mixture incorporated with OSA possess good strength and excellent environmental durability [7,8]. Moreover, the stabilization of problematic soils with OSA results in cheap disposal costs and provides good subgrade filling materials for practical engineering [9]. Turner earlier investigated the viability of using stabilized soils by OSA as engineering fillings. The compaction, unconfined strength, and resilient modulus of the stabilized soils met the requirement of highway construction, and the modified effect can be further improved when added by limestone [10]. Sharo et al. [9] evaluated the possible use of OSA as a stabilizing agent for Irbid expansive soils. The results stated that OSA was effective to increase the strength and texture by reducing the plasticity index of treated soils. Similar conclusions were obtained by Mymrin and Ponte [11].

Fly ash (FA) is well-known as the byproduct of coal combustion [12], which may threaten the human health and environment as nanoparticles [13]. Since the active component of FA can be hydrated to improve the strength and durability of materials, FA has been the most common additives used in soil stabilization [14–16]. It was found that utilizing FA to modify fine-grained soils and expansive clay soil could cause beneficial changes, such as the remarkable increase of failure stress, California Bearing Ratio (CBR) value, and the reduction of plasticity [17,18]. The FA modified soils had been proven to be effective and beneficial in embankment construction [17,19].

The described studies have proven the great prospects for the application of OSA and FA in subgrade engineering. As such, our research group proposed to modify problematic silty clay by OSA and FA (fly ash) as subgrade filling to dispose the large amounts of accumulated OSA wastes in Huadian and Fuyu of Jilin Province, China. The novel material showed excellent CBR, acceptable static and dynamic mechanical properties, promising thermal insulation capacity, and non-environmental pollution [20–22]. However, previous studies mostly focused on the physical and mechanical properties of stabilized soils by OSA with a small number of cycle loads. There is a lack of research on the deformation characteristics after long-term dynamic loadings. As the foundation of road, subgrade may be subjected to tens of thousands of long-term cyclic traffic loads which results in vibration-excited subsidence. Excessive subsidence not only affects the service capacity of road structures, but also threatens the safety of drivers. For the proposed subgrade filling (silty clay modified by OSA and FA), no studies or projects have estimated its subsidence after long-term cycle loads. Accordingly, it is necessary to investigate the deformation of this subgrade filling after a large number of cycle loads, which provides valuable guidance for the design of subgrade in a seasonally frozen region.

The OSA stabilized subgrade material proposed by our research group is intended for local use in Jilin Province, which belongs to the typical seasonally frozen region. The research on the deformation characteristics of frozen soils after traffic loads is more complicated than unfrozen soils, because subgrade in seasonally frozen regions must be subjected to freeze-thaw (F-T) cycles, otherwise causing negative effects such as water migration, frost heave, thaw settlement, and stress

redistribution on the life-cycle performance of subgrade [23]. Furthermore, the above effects can lead to pavement damage like longitudinal and transverse cracks, dislocation of cement pavement slabs, breaking in the corner of pavement slabs, and uneven settlement of the pavement slab, and so on [24,25]. In conclusion, for subgrade in seasonally frozen regions, the effect of F-T cycles on properties of materials and structures cannot be ignored.

For better understanding the deformation characteristics of using silty clay modified by OSA and FA as the subgrade material in seasonally frozen regions and thus provide valuable guidance for its practical application, this paper aims to: (1) Present the change characteristics on the total axial strain and cumulative plastic strain of modified silty clay (MSC); (2) discuss the effects of dynamic stress ratio, confining pressure, loading frequency, and F-T cycles on the cumulative plastic strain of MSC, present the cumulative plastic strain equation corresponding to the above conditions; (3) obtain the predicted values on the cumulative plastic strain of MSC, and compare with national standards to confirm the excellent performance of MSC used as the subgrade materials.

## 2. Materials

### 2.1. Raw Materials

For implementing the experimental research on the effect of F-T cycles on the deformation characteristics of MSC, raw materials including OSA, FA, and SC were obtained from Jilin province, Northeast China. Their characterization of X-ray diffraction (XRD) and Fourier transform infrared spectroscopy were measured at Testing Center of Jilin University to obtain their chemical compositions and mineral compositions.

The SC was obtained from the foundation soil of Nanguan district, Changchun, Jilin province. It is a typical subgrade material in Northeast China. Its strength and loading capacity decrease significantly after soaking water. Its plastic limit is 24.3%, liquid limit is 37.3%, and plasticity is 13.0%. Its content of $Na_2O$ is 1.99%, $K_2O$ is 23.07%, MgO is 1.32%, $K_2O$ is 3.07%, CaO is 1.25%, $Fe_2O_3$ is 4.13%, $Al_2O_3$ is 14.53%, $SiO_2$ is 68.76%, and its loss on ignition is 6.68%. The optimum moisture content is 12.2% and maximum dry density is 1.93 g/cm$^3$. The mineral compositions of SC are listed in Table 1.

**Table 1.** The mineral compositions of raw materials.

| Samples | The Mineral Compositions | | | | |
|---------|--------|-----------|--------------------|----------|----------|
| | **Quartz** | **Anorthose** | **Potassium Feldspar** | **Analcime** | **Calcitum** |
| SC | 50% | 12% | 2% | / | / |
| FA | 30% | / | / | / | / |
| OSA | 25% | 8% | / | 5% | 10% |
| | Kaolinite | Illite/montmorillonite | Mullite | Organic matter | Non-crystalline |
| SC | 5% | 30% | / | 1% | / |
| FA | / | / | 10% | / | 60% |
| OSA | / | 48% | / | 4% | / |

Notes: (1) OSA = oil shale ash; (2) FA = fly ash; (3) SC = silty clay; (4) "/" means that the mineral content is too small to be detected.

The OSA used in this study was obtained from oil shale residues which was provided by Wangqing Oil Shale Industry Park. The OSA is granular and black, dominated by inorganic compounds with a certain content of carbon and a small amount of sulfide; its content of $Na_2O$ is 2.31%, $K_2O$ is 1.84%, MgO is 2.49%, $SO_3$ is 3.29%, CaO is 6.42%, $Fe_2O_3$ is 7.22%, $Al_2O_3$ is 13.44%, $SiO_2$ is 56.28%. Its loss on ignition is 6.68%, and grain density is 1.81 g/cm$^3$. The mineral compositions of OSA are listed in Table 1. The FA used in this study was obtained from the Second Power Plant of Changchun, Jilin province. Its classification is Class F Grade 1 [26]. The FA has a high content of $SiO_2$ + $Al_2O_3$ + $Fe_2O_3$ (88.64%) and other compounds including $SO_3$ (0.24%), CaO (0.92%), and Mg, Ti, Na, K compounds (5.83%); its loss on ignition is 4.08% and grain density is 1.93 g/cm$^3$. The mineral compositions of FA are listed in Table 1.

The micro-structure figures of three raw materials obtained from SEM (scanning electron microscope) are shown in Figures 1–3.

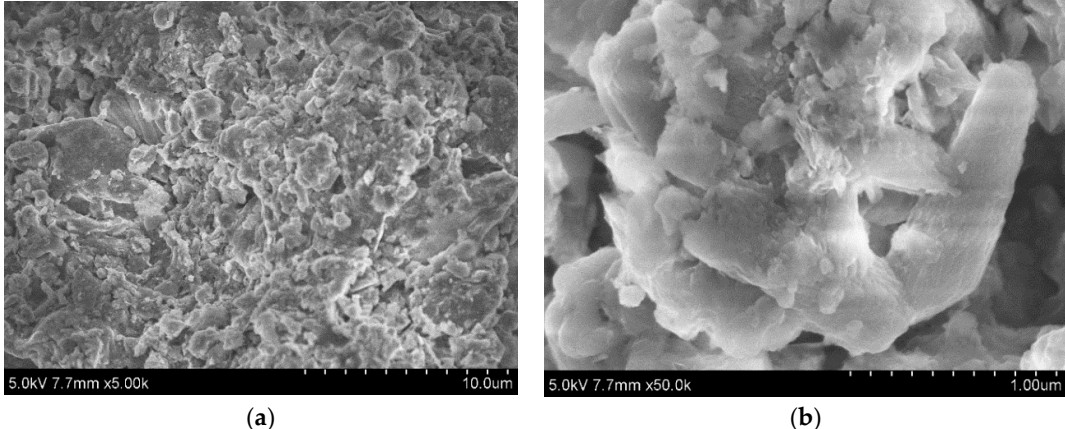

(**a**)　　　　　　　　　　　　　　　　　　(**b**)

**Figure 1.** Micro-structures of OSA. (**a**) 5 k×; (**b**) 50 k×.

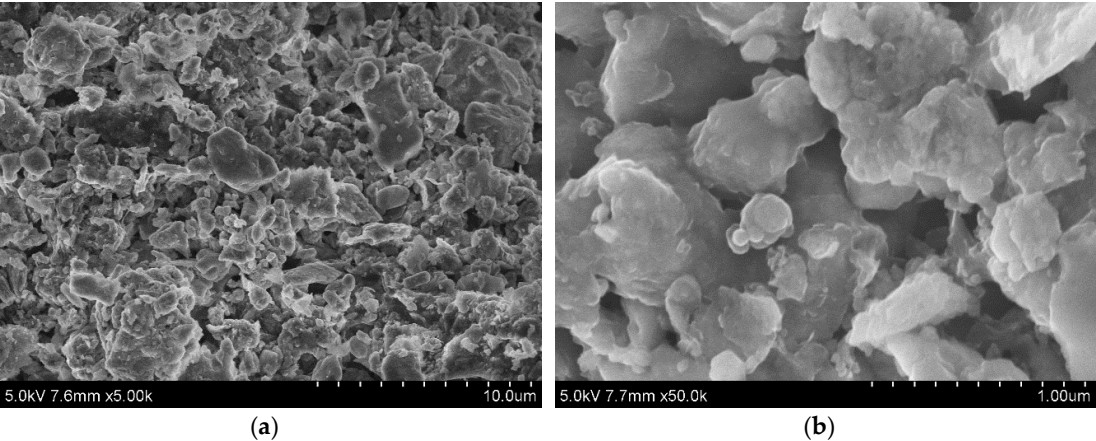

(**a**)　　　　　　　　　　　　　　　　　　(**b**)

**Figure 2.** Micro-structures of FA. (**a**) 5 k×; (**b**) 50 k×.

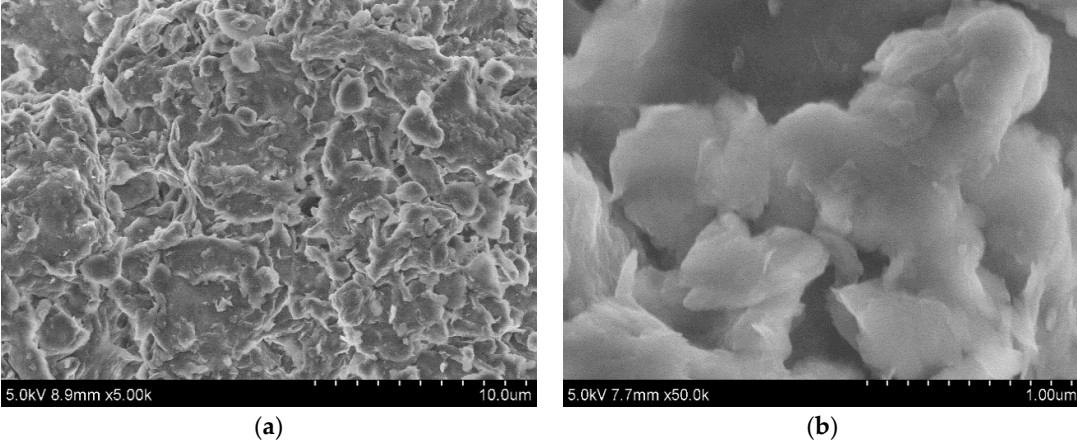

(**a**)　　　　　　　　　　　　　　　　　　(**b**)

**Figure 3.** Micro-structures of SC. (**a**) 5 k×; (**b**) 50 k×.

As shown in Figures 1–3, in the magnification of 5 k× and 50 k×, the basic structure of OSA and FA is the accumulation of many particles, which are regular shapes but different sizes; however, the microstructure of SC is a whole platy structure with layered structures, and they are closely connected in parallel or at an angle, and are stable structures. In the three raw materials, the particles of OSA and FA are smaller for the micron grade, and there are many pores in their interior. In

comparison to the OSA and SC, the particles of SC are larger and its surface is smooth. The micro-structure analysis shows that the OSA and FA possess smaller particle sizes and more pores. Previous studies [27,28] showed that smaller particle sizes and higher porosity lead to greater specific surface area and adsorption capacity. So, the OSA and FA have greater specific surface area and adsorption capacity than SC. Therefore, they might improve the physical properties of SC like adsorption capacity and strength.

## 2.2. Preparation of Mixed Samples and Testing Samples

Based on the existing research, the dry mass ratio of 2:1:2 for OSA/FA/SC and moisture content of 12.95% are selected to manufacture MSC because, through a series of conventional physical and mechanical property tests conducted by the authors, it was found that, under this ratio, the MSC has great shear strength, the smallest plasticity and the highest CBR value [20,21,29]. Table 2 lists the physical properties of MSC obtained from Refs [20,21,29]. The dry mass ratio of 2:1:2 for OSA/FA/SC and moisture content of 12.95% used in this study can be referred to Refs [20,21,29]. Before manufacturing MSC, there are three steps. First, the OSA, FA, and SC are dried in a drying oven at 105–110 °C for 24 h and then were cooled in a desiccator to room temperature. Second, the OSA, FA, and SC were mixed in a mass ratio of 2:1:2, and pure water was added with the moisture content of 12.95% by several times to obtain mixed samples with the target moisture content. This mixing process is achieved by mixing manually, and after the mixing process, the mixed samples must be powder-like. Finally, the mixed samples were placed in a humidor to let moisture to diffuse into the samples evenly.

The testing sample preparation refers to Ref [30], which stipulated that the ratio of height to diameter of triaxial test samples is 2.0~2.5, thus in this study, the diameter of samples is 39.1 mm and height is 80 mm, and in order to study the deformation characteristics of MSC in the condition of actual construction, the testing samples are manufactured with the maximum dry density (1.66 g/cm$^3$) and optimum moisture content (12.95%).

**Table 2.** The physical properties of modified SC (MSC).

| Physical Properties | OSA: FA: SC | | | | | Reference |
|---|---|---|---|---|---|---|
| | 4:3:3 | 3:4:3 | 7:5:8 | 2:1:2 | 9:3:8 | |
| Plastic Limit (%) | 27.96 | 26.94 | 27.54 | 20.20 | 25.77 | |
| Liquid Limit (%) | 40.51 | 40.17 | 42.70 | 32.6 | 42.60 | Wei et al. [20] Wei et al. [21] |
| Plasticity (%) | 12.55 | 13.23 | 15.16 | 12.4 | 16.83 | |
| CBR Value soaking for 96 h (%) | 17.00 | 13.00 | 31.00 | 40.00 | 33.00 | Cai et al. [29] |

## 3. Methods of Testing

### 3.1. Testing Scheme

The GDS dynamic triaxial testing system, made by GDS (Global Digital System) Instruments Company in England, is used in the study. Before test, MSC samples need to be consolidated with a consolidation ratio of 1.9, which is the empirical consolidation ratio calculated by the static strength of MSC. After consolidation, the stiffness of testing samples is set for 10 because the testing indicates that, when the stiffness of testing samples is set for 10, the testing samples of MSC can reach the stress design from GDS.

In order to make the testing results reflect the true performance of subgrade in the period of road service, cycle loads in the form of compression sinusoidal wave are used to simulate the loading state of MSC under vehicle-mounted load.

The loading mode in this study is stress control. A previous study showed that the loading frequency of 1 Hz corresponds to the vehicle speed of 40~50 km/h, and 2 Hz corresponds to 80~100 km/h [31]. According to the actual service speed of urban roads and expressways, the loading frequency is set as 1–2 Hz, and the cyclic vibration number is 8000. In this study, to investigate the

deformation characteristics of MSC used as subgrade materials, two set values of confining pressure 100 kPa and 200 kPa, four different dynamic stress ratios ($\sigma_d/2\sigma_c$), and different numbers of F-T cycles were used. For the above loading parameters, they were decided by the actual stress condition of subgrade construction, and were used by some related studies [32,33]. The testing scheme on deformation characteristics of MSC is listed in Table 3.

**Table 3.** Testing scheme on deformation characteristics of MSC.

| No. | Number of F-T Cycles | Confining Pressure/kPa | Axial Stress/kPa | Stress Ratio | Loading Frequency/Hz |
|-----|--------------------|----------------------|------------------|--------------|---------------------|
| 1# |   | 100 | 40 | 0.2 | 1 |
| 2# |   | 100 | 100 | 0.5 | 1 |
| 3# | 0 | 100 | 300 | 1.5 | 1 |
| 4# |   | 100 | 300 | 1.5 | 2 |
| 5# |   | 100 | 500 | 2.5 | 1 |
| 6# |   | 200 | 200 | 0.5 | 1 |
| 7# |   | 100 | 100 | 0.5 | 1 |
| 8# |   | 100 | 300 | 1.5 | 1 |
| 9# | 1 | 100 | 300 | 1.5 | 2 |
| 10# |   | 100 | 500 | 2.5 | 1 |
| 11# |   | 200 | 200 | 0.5 | 1 |
| 12# |   | 100 | 100 | 0.5 | 1 |
| 13# | 5 | 100 | 300 | 1.5 | 1 |
| 14# |   | 100 | 500 | 2.5 | 1 |

Notes: MSC = silty clay modified by oil shale ash and fly ash.

### 3.2. Data Extraction Criterion

The typical deformation curves of testing samples under cycle loads are shown in Figure 4. As shown in Figure 4, A0 is the start point of cycle loads, A1 and A2 are the regression point after each cycle load, and the vertical distances from A0 to A1 and A0 to A2 are regarded as permanent axial strain or cumulative plastic strain. C0, C1, and C2 are the maximum stress point and are also the total axial strain point for each cycle load. The vertical distances from C0 to A1 and C1 to A2 are the elastic deformation for each cycle load, which is defined as the rebounded strain. B0, B1, and B2 are the points representing semi-maximum dynamic stress in the cycle load. According to this load form in this study, they are also viewed as the equilibrium point of dynamic stress, and the vertical distances from them to A0 could also be considered as permanent axial strain or cumulative plastic strain. Substantially, the selection of A or B to analyze the cumulative plastic strain (permanent axial strain) has low impact on the analysis of the deformation characteristics of testing samples; in this study, we choose A as the plastic strain point to analyze the deformation characteristics of testing samples.

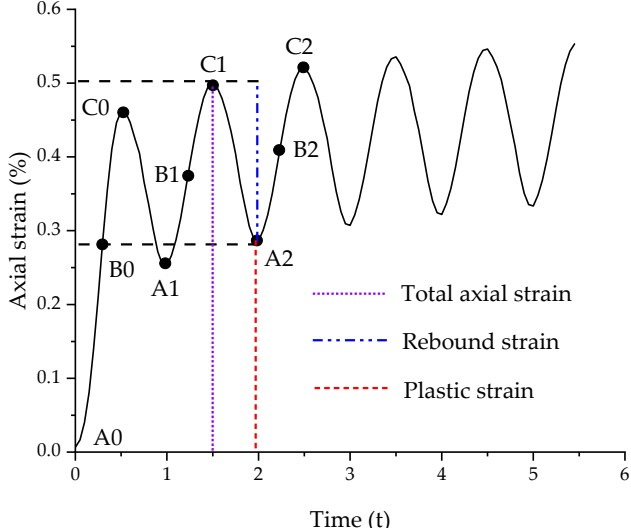

**Figure 4.** Schematic diagram on deformation curve of cycle loads.

### 3.3. Failure Criterion

During the cycle load process, it is terminated when the following phenomena occur: (1) The total axial strain of testing samples leads to 5% change of their height, indicating that the testing samples are invalid; (2) when the number of cycle loads is reached; (3) the experiment apparatus cannot output the setting value of loading stress or stress amplitude; (4) the obvious damage occurs in deformation zone of samples.

## 4. Results and Discussion

### 4.1. Shake-Down Analysis on Deformation of MSC

Stress–strain is the relation between loading stress and deformation of testing samples; it can effectively reflect the elastoplastic, brittle, and yield failure characteristics of testing samples. Taking the stress–strain curves of MSC with confining pressure of 100 kPa and axial stress of 40 kPa, 100 kPa, and 300 kPa, respectively, which don't undergo F-T cycles, as the representative, the stress–strain curve characteristics of MSC under cycle loads are shown in Figure 5.

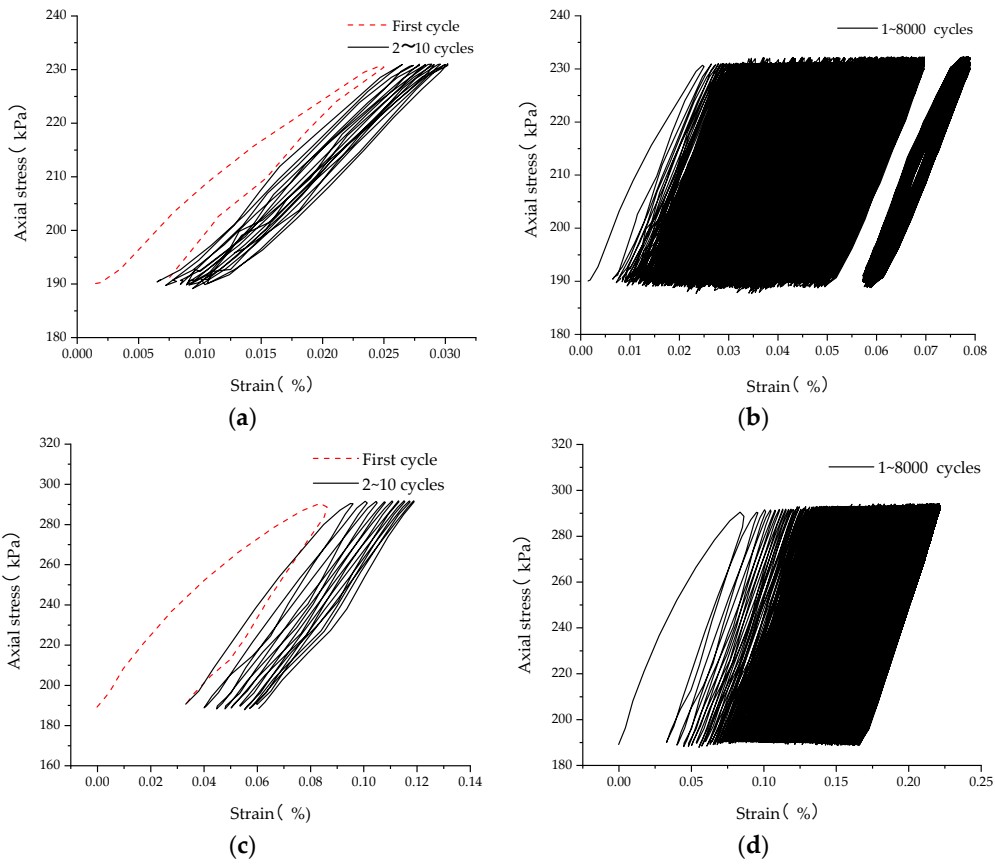

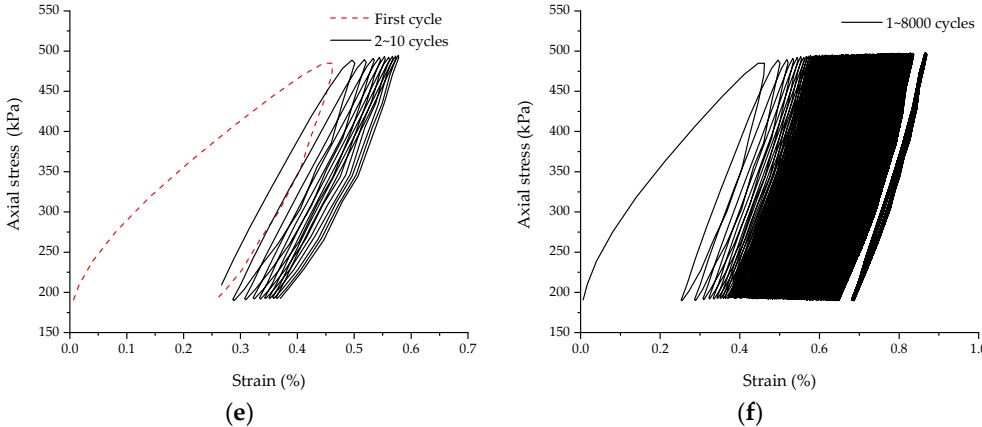

**Figure 5.** The stress–strain curves on the experiment of cycle loads for MSC. (**a**) 10 cycles ($\sigma_c$ = 100 kPa, $\sigma_d$ = 40 kPa); (**b**) 8000 cycles ($\sigma_c$ = 100 kPa, $\sigma_d$ = 40 kPa); (**c**) 10 cycles ($\sigma_c$ = 100 kPa, $\sigma_d$ = 100 kPa); (**d**) 8000 cycles ($\sigma_c$ = 100 kPa, $\sigma_d$ = 100 kPa); (**e**) 10 cycles ($\sigma_c$ = 100 kPa, $\sigma_d$ = 300 kPa); (**f**) 8000 cycles ($\sigma_c$ = 100 kPa, $\sigma_d$ = 300 kPa).

As shown in Figure 5, the stress–strain curve is described by loading cycles of 1 to 10 and 1 to 8000. The deformation trend of MSC conforms to the plastic shake down theory [34] for the following reasons: In the process of cyclic loading, MSC shows accumulation of plastic strain obviously; the stress–strain relationship shows a hysteretic property; MSC absorbs energy from the cyclic loads, its stress-strain relationship tends to be stable, and its plastic strain increases and tends to be stable in a certain rule. In this section, for MSC without undergoing F-T cycles, the stress–strain curves are analyzed in three loading conditions. Results indicating that MSC, which have not been frozen and thawed, accords with the plastic shake down theory. In addition, the results show MSC undergoing 1 and 5 F-T cycles are similar to those of MSC going through 0 F-T cycle in this work. In three stress states, the stress–strain curve shows an obvious hysteretic loop, and with the increases of load number, the quantity of hysteretic loop increases and its growth ratio of deformation tends to be stable. In the axial stress of 40 kPa, 100 kPa, and 300 kPa, the cumulative strain ratio of axial deformation in the first 10 cycle load is extremely large, especially after the first cycle load. For the axial stress of 40 kPa, the cumulative strain after the first cycle load was 0.01892%, accounting for 29.7% of the cumulative strain after 8000 cycles. For the axial stress of 100 kPa, after the first load cycle, the cumulative strain was 0.0716%, accounting for 30% of the cumulative strain after 8000 cycle loads. When the axial stress was 300 kPa, the cumulative strain after the first cycle load was 0.2226%, accounting for 33.7% of the cumulative strain after 8000 cycle loads.

In order to further explore the relation between the axial deformation of MSC and cycle loads, 3 MSC samples were selected to obtain the relationship diagram between axial deformation and cycle load number, as shown in Figure 6. The variations on the axial strain of three samples with load number are shown in colored ribbons, and the area covered by colored ribbons is the change path of axial deformation; under the same confining pressure, the larger the axial stress is, the larger the total axial strain and cumulative plastic strain value are, and the wider the ribbon area is. The value on the top contour of ribbon area is the total axial strain of each cycle load. The value on the bottom contour of ribbon area represents the cumulative plastic strain of each cycle load. The width of ribbon area is the rebound strain of each cycle load. As shown in Figure 6, the total axial strain and cumulative plastic strain increase as the number of cycle load increases; especially, under the low number of cycle loads, the total axial strain increases significantly with the increase of cycle load number, and when the load number increases gradually, the increase trend of axial strain slows down and tends to be stable. However, the rebound strain increases slightly with the increase of load number, but the change trend is not obvious.

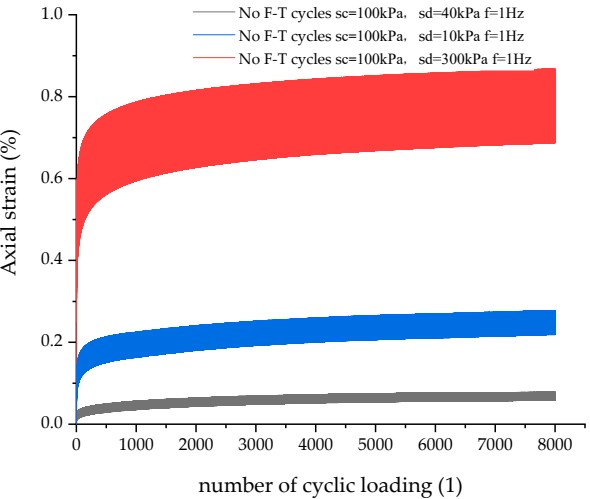

**Figure 6.** The relation between the axial strain and number of cycle loads for MSC.

*4.2. Axial Strain Characteristics of MSC Undergoing 0~8000 Cycle Loads*

In order to analyze the axial strain characteristics of MSC under different numbers of cyclic loads and F-T cycles, the axial strain of MSC, undergoing 0~8000 cycle loads and 0, 1, and 5 F-T cycles, is shown in Figure 7. The deformation characteristics of MSC are analyzed by the cumulative plastic strain, because the plastic cumulative strain is often used to evaluate the deformation characteristics of railway and highway subgrade under long-term traffic load, which is an important parameter that cannot be ignored in practical engineering.

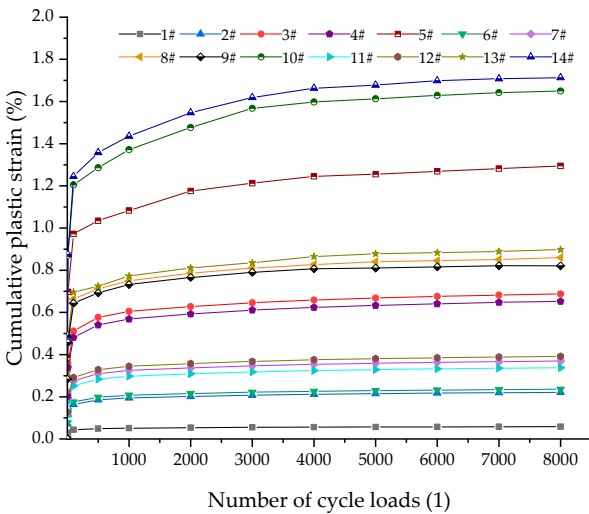

**Figure 7.** Relationship curve between the cumulative plastic strain and number of cycle loads for MSC.

As shown in Figure 7, the cumulative plastic strain first increases and then tend to be stable with the increases of cycle load numbers. The minimum axial strain appeared in the Sample #1 (Table 1), undergoing 0 F-T cycles and applied by the confining pressure of 100 kPa and axial stress of 40 kPa, and its axial strain increased linearly with the increase of the number of cycle loads; the maximum axial strain appeared in the Sample #14, undergoing 5 F-T cycles and applied by the confining pressure of 100 kPa and axial stress of 500 kPa. The cumulative plastic strain of testing samples of MSC is very small after 8000 cycle loads, and even undergoing 5 F-T cycles, it is still lower than 2%, and no damage caused by strain overloading occurs, and thus compared with previous studies [35,36] about deformation characteristics in soil, which takes the axial strain of 10%~15% as the soil

failure criterion, the deformation characteristics indicate that MSC possesses better stability performance.

It is worth noting that the cumulative plastic strain of MSC accumulates rapidly in the initial stage of cycle loads, contributing to most of the deformation of the cumulative plastic strain after large number of cycle loads, and in order to quantify the difference of cumulative plastic strain between the initial stage of cycle loads and 8000 cycle loads, the cumulative plastic strain of MSC, after 1, 10, and 100 cycle loads, is summarized and compared with that of 8000 cycle loads; results are shown in Table 4. As shown in Table 4, the axial plastic strain in the initial 100 cycle loads occupies extremely high proportion of final axial cumulative plastic strain occurring after 8000 cycle loads, showing the importance of axial plastic strain that occurs in the early stage of cycle loads. The cumulative plastic strain of MSC, after 1, 10, and 100 cycle loads, occupies for 28.72%~35.31%, 49.86%~55.59%, and 70.87%~78.39% of final axial cumulative plastic strain occurring after 8000 cycle loads, indicating that testing samples show remarkable plastic stability in the first 100 cycles of cycle loads. Dynamic stress ratio, confining pressure, loading frequency, and F-T cycles will have some extent of impact on the cumulative plastic strain of MSC after 8000 cycle loads (Figure 7), but these factors have insignificant impact on the axial plastic strain in the early stage of cycle loads (1, 10, and 100 cyclic loads). Under different dynamic stress ratios, taking the ratio of the cumulative plastic strain after the first cycle load to that of occurring after 8000 cycle loads as an example, when the stress ratio is 0.2, the proportion is 29.65%. When the stress ratio is 0.5, the proportion is 29.65%~32.58%. When the stress ratio is 1.5, the proportion is 33.69%~35.31%. When the stress ratio is 2.5, the proportion is 28.72%~29.81%. The above analysis shows that the plastic strain values of MSC under the four stress ratios are close to each other after the first cycle. In Table 4, it is obvious that the above analysis and result are also applicable to the analyses of confining pressure, loading frequency, and F-T cycles.

**Table 4.** The ratio of cumulative plastic strain after 1, 10, and 100 load cycles to that after 8000 load cycles for MSC.

| Dynamic Stress Ratio | F-T Cycle Number | Confining Pressure (kPa) | Frequency (Hz) | R after 1 Load Cycles | R after 10 Load Cycles | R after 100 Load Cycles |
|---|---|---|---|---|---|---|
| 0.2 | 0 | 100 | 1 | 29.65% | 50.25% | 72.26% |
| 0.5 | 0 | 100 | 1 | 29.65% | 49.86% | 70.87% |
| 0.5 | 0 | 200 | 1 | 32.38% | 52.12% | 74.37% |
| 0.5 | 1 | 100 | 1 | 32.06% | 51.46% | 72.95% |
| 0.5 | 1 | 200 | 1 | 31.26% | 50.89% | 72.38% |
| 0.5 | 5 | 100 | 1 | 32.58% | 52.12% | 74.37% |
| 1.5 | 0 | 100 | 1 | 33.69% | 52.12% | 74.37% |
| 1.5 | 0 | 100 | 2 | 34.36% | 51.85% | 73.76% |
| 1.5 | 1 | 100 | 1 | 35.31% | 55.59% | 77.16% |
| 1.5 | 1 | 100 | 2 | 34.55% | 53.87% | 78.39% |
| 1.5 | 5 | 100 | 1 | 35.16% | 54.58% | 77.47% |
| 2.5 | 0 | 100 | 1 | 29.81% | 53.90% | 75.14% |
| 2.5 | 1 | 100 | 1 | 29.39% | 52.79% | 73.03% |
| 2.5 | 5 | 100 | 1 | 28.72% | 51.14% | 72.68% |

Notes: R = the ratio of cumulative plastic strain to that of 8000 cycle loads.

### 4.3. Plastic Strain Characteristics of MSC after 8000 Cyclic Loads

Previous studies showed that confining pressure, axial stress, loading frequency, F-T cycles, and other factors all affect the axial deformation characteristics of soil materials after cycle loads [37–39]. Axial deformation of soil samples, especially cumulative plastic strain, is a very important parameter in geotechnical engineering [40]. In this section, the cumulative plastic strain after 8000 cycle loads is taken as the goal to discuss the effect of confining pressure, axial stress, loading frequency, and F-T cycles on axial deformation of MSC. In the following interpretation, the cumulative plastic strain after 8000 cycle loads will be referred to as ultimate cumulative plastic strain for short. Figure 8 shows the

relationship between the ultimate cumulative plastic strain and dynamic stress of MSC. In Figure 8, the four dynamic stress ratios $r$ ($r = \sigma_d/2\sigma_c$, axial stress for $\sigma_d$ and confining pressure for $\sigma_c$) were 0.2, 0.5, 1.5, and 2.5, respectively. It is obvious that the ultimate cumulative plastic strain of MSC increases linearly with the increase of dynamic stress ratio, and with the progress of freezing-thawing cycles, the linear growth trend is still valid, indicating that F-T cycles will increase the ultimate cumulative strain in the same dynamic ratio.

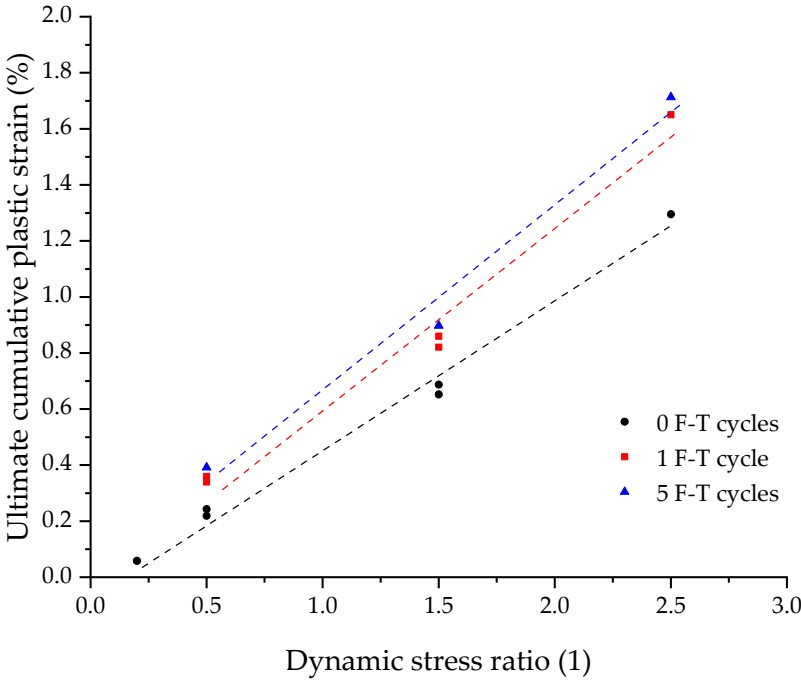

**Figure 8.** The relation between the ultimate cumulative plastic strain and dynamic stress ratio of MSC.

Figure 9a shows the relation between the ultimate cumulative plastic strain and confining pressure of Samples #2, #6, #7, and #11, which the dynamic stress is equal to 0.5. Before F-T cycle, with the confining pressure increasing from 100 kPa to 200 kPa, the ultimate cumulative plastic strain increases from 0.22% to 0.24%, and after one freezing-thawing cycle, the ultimate cumulative plastic strain of MSC decreases from 0.36% to 0.34%. It can be concluded that, under low confining pressure, the ultimate cumulative strain of MSC after multi-cycle loads does not change significantly with the change of confining pressure; that is, under low confining pressure, it is invalid to reduce the cumulative plastic strain of MSC after multi-cycle loads by increasing the confining pressure. It is worth noting that this conclusion is only applicable to low confining pressures, because also to the confining pressures of 100 and 200 kPa that are discussed in this manuscript. Figure 9b shows the relation between the ultimate cumulative plastic strain and the loading frequency of Samples #3, #4, #8, and #9, where the confining pressure is 100 kPa and axial stress is 300 kPa. No matter the number of F-T cycle is 0 or 1, the ultimate axial cumulative plastic strain of MSC decreases with the increase of frequency, indicating that increasing the vehicle speed is beneficial to the reduction of the ultimate cumulative plastic strain of MSC.

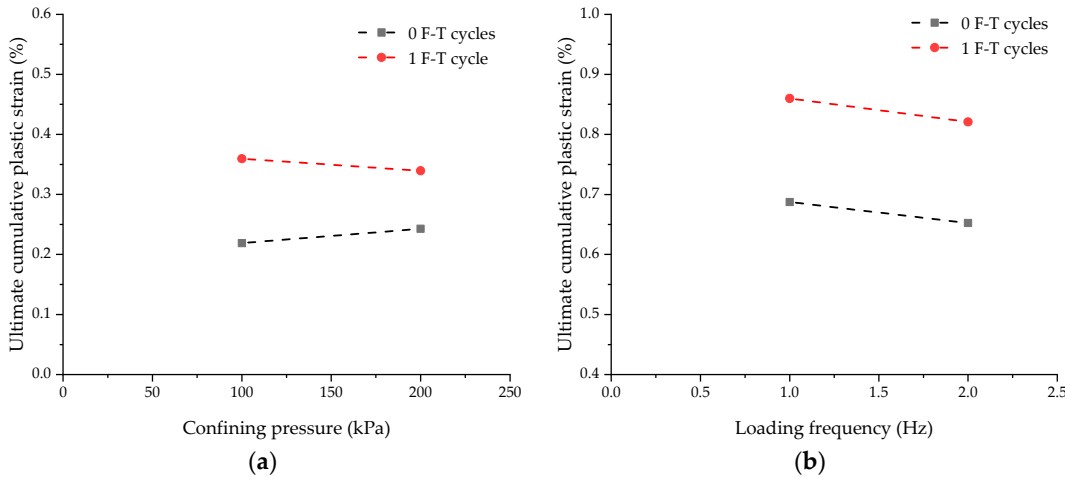

**Figure 9.** The relation between the ultimate cumulative plastic strain, confining pressure, and loading frequency of MSC. (**a**) Confining pressure (kPa); (**b**) Loading frequency (Hz).

In addition to the stress, the F-T cycle is another important factor for the ultimate cumulative plastic strain of MSC. Based on the conditions including the load frequency of 1 Hz, confining pressure of 100 kPa, 8000 cycle loads, axial stress of 100, 300, and 500 kPa, the MSC samples after 0, 1, and 5 F-T cycles are utilized to analyze the effect of F-T cycles on the ultimate cumulative plastic strain of MSC. Figure 10 shows the relation between the ultimate cumulative plastic strain and F-T cycles of MSC, respectively. After 1 F-T cycle, the ultimate cumulative plastic strain of MSC increases rapidly, and the larger axial stress, the larger cumulative plastic strain increment after 1 freeze-thaw cycle. After 5 F-T cycles, the ultimate cumulative plastic strain of MSC is only slightly higher than that after 1 F-T cycle. The above analysis shows that F-T cycles will have an impact on the cumulative plastic strain of MSC, but after the large deformation of the first F-T cycle, it gradually becomes stable.

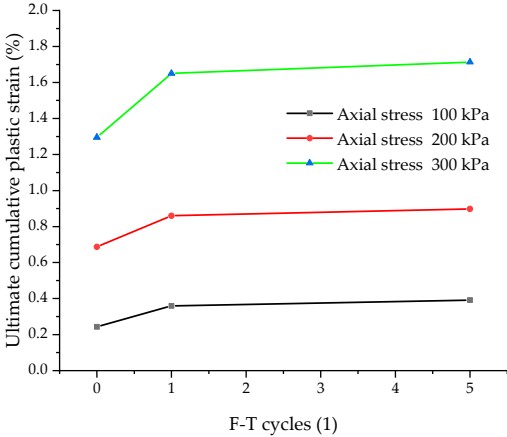

**Figure 10.** The relation between the ultimate cumulative plastic strain and F-T cycles of MSC.

*4.4. Prediction Model on the Cumulative Plastic Strain of MSC*

Based on the deformation characteristics of MSC under multiple cycle loads, an appropriate model is proposed that is used to predict the settlement characteristics of subgrade using MSC as the filling material. In previous studies [41,42], the cumulative plastic strain of soil medium can be expressed by the power function as Equation (1):

$$\varepsilon_p = AN^b \tag{1}$$

where $\mathcal{E}_p$ is cumulative plastic strain, $N$ is the number of cycle loads, $A$ and $b$ are material parameters.

### 4.4.1. Normalized Prediction Model

In order to discuss the variation trend on deformation of MSC with cycle loads, the normalization method is used for analysis; point clusters are obtained by normalizing the cumulative plastic strain of MSC after 100 cycle loads, as shown in Figure 11, which can be described by the following expression:

$$\frac{\mathcal{E}_p}{\mathcal{E}_{100}} = AN^b \tag{2}$$

where $\mathcal{E}_{100}$ is the cumulative plastic strain of MSC after 100 cycle loads; the fitting curve is shown in Figure 11, and the fitting results are $A = 0.5642$, $b = 0.0994$, $R^2 = 0.949$, and thus the Equation (2) can be converted into Equation (3):

$$\mathcal{E}_p = \mathcal{E}_{100} 0.5642 N^{0.0994} \tag{3}$$

The cumulative plastic strain $\mathcal{E}_{100}$ of 14 samples listed in Table 1 after 100 cycle loads was calculated, and based on the calculated results, the point diagram of $\mathcal{E}_{100}$, with the change of dynamic stress ratio, is drawn, and the linear fitting is carried out; the results are shown in Figure 12. In Figure 12, the slope of fit line, corresponding to 0, 1, and 5 F-T cycles, represents the sensitivity of the cumulative plastic strain and dynamic stress ratio of MSC after 100 cyclic loads, and it is obvious that $\mathcal{E}_{100}$ of after F-T cycles is more sensitive to the change of dynamic stress ratio than that which does not undergo F-T cycles.

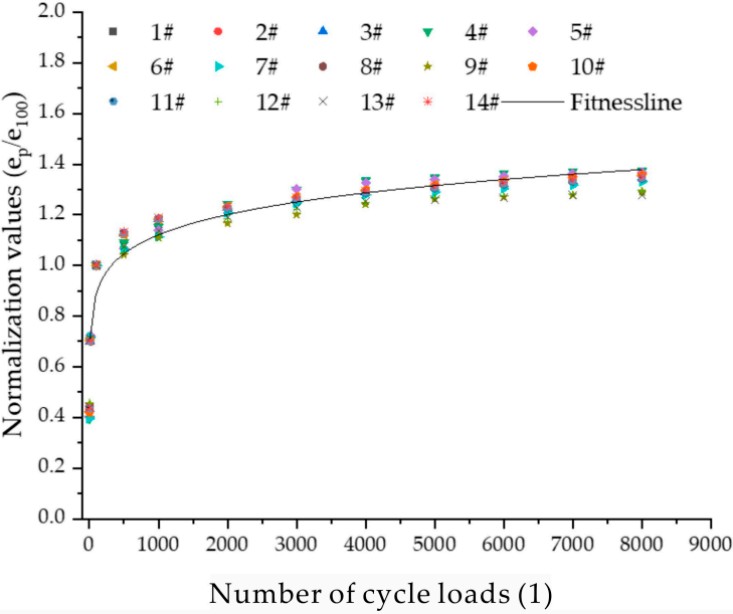

**Figure 11.** Normalization result on the cumulative plastic strain of MSC.

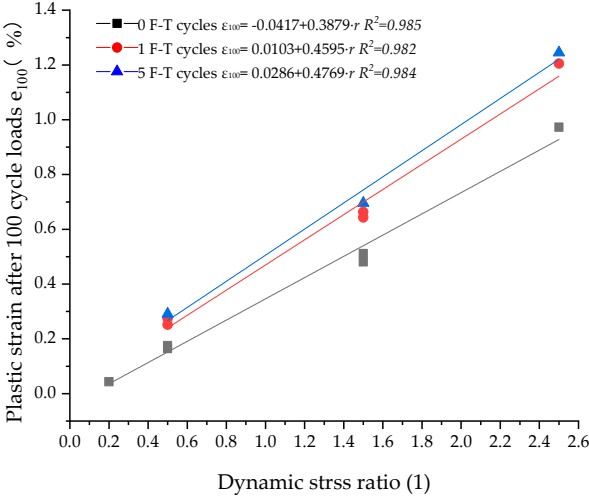

**Figure 12.** The relation between the cumulative plastic strain and dynamic stress ratio of MSC after 100 cycle loads.

### 4.4.2. Logarithmic Prediction Model

Barksdale [43] conducted research on the strain characteristic of granular material with large number of cycle loads. Results indicate that, based on the base-10 logarithm, the cumulative plastic strain and number of cycle loads can be described by a near-linear relationship; the value of $\log A$, where the meaning of A have been shown in Equation (1), is related to the cumulative plastic strain after the first cycle load. Therefore, the relation between the cumulative plastic strain and number of cycle load of MSC can be expressed as follows:

$$\varepsilon_p = \varepsilon_1 + b \log N \tag{4}$$

where $\varepsilon_p$ is the cumulative plastic strain after a large number of cycle roads, and $\varepsilon_1$ is the cumulative plastic strain after the first dynamic cycle loads. In order to make Equation (4) suitable to predict the cumulative plastic strain of MSC after a large number of cycle loads, Figure 13 plots the relationship between the cumulative plastic strain of MSC and the cycle load number. In Figure 13, there is an obvious linear relationship between the cumulative plastic strain and cycle load number, especially when the stress ratio is low. When the stress ratio is 0.2 and 0.5, the value of accumulative plastic strain is approximately a flat line, and when the stress ratio increases, so does the slope of the line. When the stress ratio is 1.5 and 2.5, the values of cumulative plastic strain is a broken line and the inflection point is corresponding to the cycle load number of 100. According to the above difference, two different functions can be used to analyze the cumulative plastic strain of MSC under a large number of cycle loads. When the stress ratio is 0.2 and 0.5, Equation (4) can be used for fitting. When the stress ratio is 1.5 and 2.5, the expression can be referred to the following equation:

$$\varepsilon_p = \varepsilon_{100} + b\left(\log N - \log 100\right) \tag{5}$$

according to Equation (4), its value is $\varepsilon_{100} = \varepsilon_1 + a \log 100$, where $a$ is a material parameter, and then Equation (5) can be converted to:

$$\varepsilon_p = \varepsilon_1 + a \log 100 + b \log\left(\frac{N}{100}\right) \tag{6}$$

In this work, under different stress conditions, the relation between the cumulative plastic strain of MSC and cycle load number, fitted by Equations (4) and (6), is shown in Table 5. As shown in Table 5 under the condition of cycle loads, the correlation coefficients of the prediction model on cumulative plastic strain of MSC are very high, indicating that the model is suitable for predicting cumulative

plastic strain of MSC. The value of material parameter *b* in the prediction model of cumulative plastic strain reflects the sensitivity of the plastic deformation of MSC under different stress conditions and F-T cycles. When the stress ratio is 1.5 and 2.5, the axial permanent strain of MSC can be expressed by subsection prediction model, and the b value for *N* > 100 is smaller than that for *N* ≤ 100, which verifies the earlier conclusion that the small number of cycle loads (the first 100 cycles) contribute greatly to the total plastic deformation.

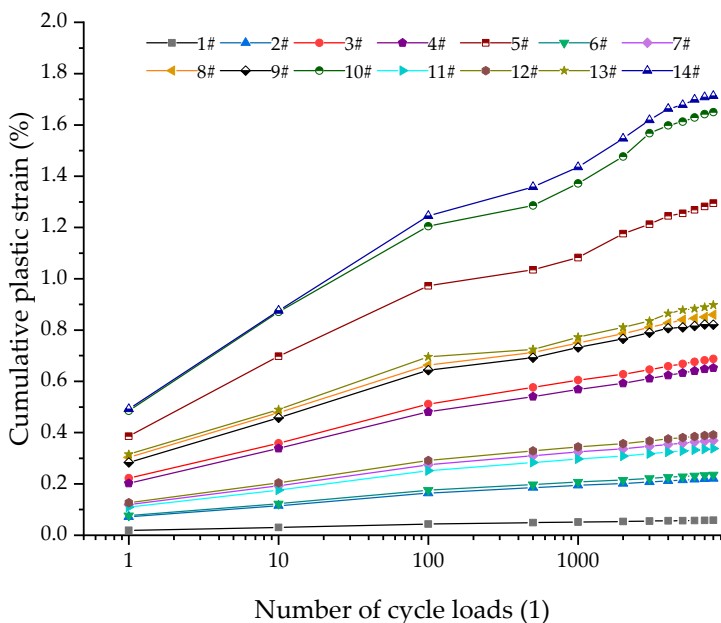

**Figure 13.** The cumulative plastic strain of MSC changes with the cycle load number.

**Table 5.** Prediction models of cumulative plastic strain of MSC after cycle loadings.

| Dynamic Stress Ratio | Confining Pressure (kPa) | F-T Cycle Number | Frequency (Hz) | Model (N: Cyclic Load Number) | Correlation Coefficient |
|---|---|---|---|---|---|
| 0.2 | 100 | 0 | 1 | $\varepsilon_p = 0.0208 + 0.00987 logN$ | 0.98 |
| 0.5 | 100 | 0 | 1 | $\varepsilon_p = 0.0786 + 0.03733 logN$ | 0.98 |
| 0.5 | 200 | 0 | 1 | $\varepsilon_p = 0.0839 + 0.03984 logN$ | 0.98 |
| 0.5 | 100 | 1 | 1 | $\varepsilon_p = 0.1314 + 0.06239 logN$ | 0.98 |
| 0.5 | 200 | 1 | 1 | $\varepsilon_p = 0.1203 + 0.05713 logN$ | 0.98 |
| 0.5 | 100 | 5 | 1 | $\varepsilon_p = 0.1394 + 0.06616 logN$ | 0.98 |
| 1.5 | 100 | 0 | 1 | $\varepsilon_p = 0.2226 + 0.14258 logN \ (N \leq 100)$ <br> $\varepsilon_p = 0.5078 + 0.09436 log\left(\dfrac{N}{100}\right)(N > 100)$ | 0.99 |
| 1.5 | 100 | 0 | 2 | $\varepsilon_p = 0.2026 + 0.13858 logN \ (N \leq 100)$ <br> $\varepsilon_p = 0.4798 + 0.08994 log\left(\dfrac{N}{100}\right)(N > 100)$ | 0.99 |
| 1.5 | 100 | 1 | 1 | $\varepsilon_p = 0.3036 + 0.17884 logN \ (N \leq 100)$ <br> $\varepsilon_p = 0.6613 + 0.10149 log\left(\dfrac{N}{100}\right)(N > 100)$ | 0.99 |
| 1.5 | 100 | 1 | 2 | $\varepsilon_p = 0.2836 + 0.17884 logN \ (N \leq 100)$ <br> $\varepsilon_p = 0.6413 + 0.09733 log\left(\dfrac{N}{100}\right)(N > 100)$ | 0.99 |

| 1.5 | 100 | 5 | 1 | $\varepsilon_p = 0.3157 + 0.18684 log N \ (N \leq 100)$ <br> $\varepsilon_p = 0.68938 + 0.10446 log\left(\dfrac{N}{100}\right)(N > 100)$ | 0.99 |
|-----|-----|---|---|---|------|
| 2.5 | 100 | 0 | 1 | $\varepsilon_p = 0.386 + 0.2972 log N \ (N \leq 100)$ <br> $\varepsilon_p = 0.9804 + 0.1570 log\left(\dfrac{N}{100}\right)(N > 100)$ | 0.99 |
| 2.5 | 100 | 1 | 1 | $\varepsilon_p = 0.485 + 0.3652 log N \ (N \leq 100)$ <br> $\varepsilon_p = 1.2154 + 0.22402 log\left(\dfrac{N}{100}\right)(N > 100)$ | 0.99 |
| 2.5 | 100 | 5 | 1 | $\varepsilon_p = 0.492 + 0.378 log N \ (N \leq 100)$ <br> $\varepsilon_p = 1.248 + 0.24440 log\left(\dfrac{N}{100}\right)(N > 100)$ | 0.99 |

*4.5. Predict the Value of Cumulative Plastic Strain after a Large Number of Cycle Loads (10 Million Times)*

For subgrade, the residual deformation can be calculated by knowing the cumulative plastic strain of subgrade material under long-term load. The equation is as follows:

$$s = \sum h_i \varepsilon_{pi} \tag{7}$$

where $S$ is the residual deformation of subgrade material; $h_i$ is the depth of subgrade influenced by cycle loads; $\varepsilon_{pi}$ is the cumulative plastic strain of subgrade materials. According to the previous studies [44–46], the depth, which represents the depth of subgrade influenced by cycle loads, changes from 0.8 m to 1.2 m, and in order to present the residual deformation in the worst case, $h_i = 1.2$ m is used to calculate the maximum residual deformation.

For highway subgrade, the specification for Design of Highway Subgrades (JTG D30-2015) [47] stipulates that the requirement of post-construction settlement in soft soil areas is as follows: the post-construction settlement of the bridge abutment, which is adjacent to the subgrade of expressway and Grade I highway, should not exceed 10 cm, and the subgrade shall not exceed 30 cm. For railway subgrade, in Code for Design of Railway Earth Structure (TB 10001-2016) [48], the post-construction settlement standard of railway subgrade is: For Grade I railway, the post-construction settlement of general location and the transition section between road and bridge should not be more than 20 cm and 10 cm, respectively; the passenger railway line, with ballast track, is 5~15 cm, while the transition section of the road and bridge is 3~8 cm.

In previous studies [33,46,49], almost all of the cycle load number, used to study permanent deformation characteristics or cumulative plastic strain of soil medium, are lower than one million times. In this manuscript, in order to present the residual deformation in more rigorous cases than previous studies, we assume 10 million times of cycle loads to predict the cumulative plastic strain of MSC used as subgrade material. According the results in Section 4.3, we know that the different levels of confining pressure have insignificant impact on the cumulative plastic strain of MSC and the less load frequency lead to the deformation of MSC, and the testing schemes on the confining pressure of 100 kPa and 1 Hz frequency are more than those of 200 kPa (Table 3) and 2 Hz, so we choose the confining pressure of 100 kPa and frequency of 1 Hz. Furthermore, to present the prediction results under more conditions, the dynamic stress ratio of 0.5, 1.5, and 2.5, and 0 and 5 F-T cycles are chosen, and the specific prediction schemes and results are shown in Table 6. As shown in Table 6, under the condition of 100 kPa confining pressure, 1 Hz, 0 and 5 F-T cycles, and dynamic stress ratios of 0.5, 1.5, and 2.5, the predicted values of the cumulative plastic strain of MSC range from 0.38 cm to 2.71 cm, and these predicted values are well within the standards of Design of Highway Subgrades (JTG D30-2015) [47] and Code for Design of Railway Earth Structure (TB 10001-2016) [48], indicating that the cumulative plastic strain of MSC is small and is suitable to be used as subgrade materials in its life cycle.

**Table 6.** The specific schemes and predicted results of MSC deformation.

| Dynamic Stress Ratio | Confining Pressure (kPa) | F-T Cycle Number | Frequency (Hz) | Model (*N*: 10 Million) | Predicted Results (cm) |
|---|---|---|---|---|---|
| 0.5 | 100 | 0 | 1 | $\varepsilon_p = 0.0786 + 0.03733 logN$ | 0.38 |
| 0.5 | 100 | 5 | 1 | $\varepsilon_p = 0.1394 + 0.06616 logN$ | 0.67 |
| 1.5 | 100 | 0 | 1 | $\varepsilon_p = 0.2226 + 0.14258 logN \ (N \leq 100)$ <br> $\varepsilon_p = 0.5078 + 0.09436 log\left(\dfrac{N}{100}\right)(N > 100)$ | 1.07 |
| 1.5 | 100 | 5 | 1 | $\varepsilon_p = 0.3157 + 0.18684 logN \ (N \leq 100)$ <br> $\varepsilon_p = 0.68938 + 0.10446 log\left(\dfrac{N}{100}\right)(N > 100)$ | 1.32 |
| 2.5 | 100 | 0 | 1 | $\varepsilon_p = 0.386 + 0.2972 logN \ (N \leq 100)$ <br> $\varepsilon_p = 0.9804 + 0.1570 log\left(\dfrac{N}{100}\right)(N > 100)$ | 1.92 |
| 2.5 | 100 | 5 | 1 | $\varepsilon_p = 0.492 + 0.378 logN \ (N \leq 100)$ <br> $\varepsilon_p = 1.248 + 0.24440 log\left(\dfrac{N}{100}\right)(N > 100)$ | 2.71 |

## 5. Conclusions

In this study, the deformation characteristic of MSC after F-T cycles were measured. The effects of dynamic stress ratio, confining pressure, loading frequency, and F-T cycles on the cumulative plastic strain of MSC were investigated. Research results lead to the following conclusions:

(1) As the number of cycle loads increases, the axial strain of MSC begins to increase rapidly, then slowly, and finally tends to be stable. The stress–strain curve of MSC conforms to the shakedown theory of materials. The cumulative plastic strain of MSC after 100 cycle loads occupies for 70.87%~78.39% of that after 8000 cycle loads, indicating that MSC possesses remarkable plastic stability after 100 cycles of cycle loads.

(2) The larger the stress ratio, the larger the axial strain value of MSC under the same number of cycle loads. Under the same low stress ratio, increasing confining pressure and loading frequency have insignificant effect on the cumulative plastic strain of MSC after 8000 loads. The F-T cycles can increase the deformation sensitivity of MSC. Moreover, before and after F-T cycles, the outcomes of dynamic stress ratio and confining pressure operate on MSC are the same.

(3) For the logarithmic prediction model, when the dynamic stress ratio is high (in this study, it is 1.5 and 2.5), there is a linear relationship between the cumulative plastic strain and piecewise function of load number, and the proposed prediction model has a high correlation coefficient with the testing data.

(4) The study shows that the cumulative plastic strain of MSC ranges from 0.38 cm to 2.71 cm. Compared with the related standards [47,48], the cumulative plastic strain of MSC is well under the thresholds, indicating that MSC is an excellent kind of material to be used as subgrade fillings.

In summary, the deformation characteristic of MSC after F-T cycles indicates that MSC possesses remarkable stability and accords with the related standards of subgrade construction. Therefore, utilizing MSC as subgrade fillings can achieve the purpose of disposing industry solid wastes, thus improving the economic and environmental sustainability of subgrade.

**Author Contributions:** Conceptualization—Q.L.; Data curation—Y.Z.; Funding acquisition—F.W. and Y.Z.; Investigation—Q.L., Y.Z., L.H. and S.H.; Methodology—Q.L. and Y.Z.; Writing – original draft—Q.L., H.W. and Z.C.; Writing – review & editing—Q.L. and H.W.

**Funding:** This work was supported by the National Key Research and Development Program of China (grant number 2018YFB1600200); National Natural Science Foundation of China (51578263); Science and Technology Project of Jilin Province Transportation Department [No. 2017ZDGC6]; Guangxi Natural Science Foundation (2018GXNSFAA281339); Science and Technology Research and New Product Prototype Project of Xingning district, Naning (2018A02); Science and Technology Base Special Project of Nanning (20185071-1).

**Conflicts of Interest**: The authors declare no conflict of interest.

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
