# Peer review of "Experimental Research on Deformation Characteristics of Using Silty Clay Modified by Oil Shale Ash and Fly Ash as the Subgrade Material after Freeze-Thaw Cycles"

_sustainability, doi:10.3390/su11185141_

Round 1

Reviewer 1 Report

This paper investigates the axial deformation of MSC with different levels of cycle load number, stress ratio, confining pressure, loading frequency. The cumulative plastic strain is studied. Normalized and logarithmic models on the plastic cumulative strain prediction of MSC, they have a high correlation coefficient with testing data. Here are several questions

In the introduction part, it does not clearly show a clear state of art. How did the authors obtain the numbers of the raw materials in page 3, including the strength and loading parameters. How did the authors define plastic limit, liquid limit and plasticity and the optimum moisture content? Pleses do more explanation. Its content of Na2O is 2.31%...6.68%, please provide the support materials. Figure 1-3 I cannot see the SEM images detailed information from the pictures including the scales, voltage, etc. I cannot make any comparisons without the size or scale in these images How did the author get the conclusions “ The micro structure analysis shows that the OSA and FA possess … adsorption capacity and strength. “please add relevant references or provide any experiments result. What is the GDS mean? Please add more information about your testing machine Conclusion section is a bit confusing. The authors should provide better explanation for this

Author Response

Responses to Reviewers’ Comments on Manuscript ID sustainability-518690

Dear Reviewer 1,

We would like to express our sincere gratitude your thoughtful comments and helpful suggestions for improving the quality of this paper. We have revised the manuscript entitled " Experimental Research on Deformation Characteristics of Using Silty Clay Modified by Oil Shale Ash and Fly Ash as the Subgrade Material after Freeze-Thaw Cycles " (sustainability-573955) according to your comments. Below are point-by-point responses to your comments. The corresponding modifications and corrections were made and highlighted in red in the revised manuscript (MS).

If there is any question regarding this version of the manuscript, please let us know. We are looking forward to receiving your evaluation.

Please you download the PDF

Best regards,

Fuyu Wang and Yangpeng Zhang

Jilin University

Changchun, China

Comment 1:

In the introduction part, it does not clearly show a clear state of art.

Response:

We have revised this introduction to be clearly show a clear state of art, by adding related references, listing the goals in this study, and simplifying the sentences in this manuscript.

Comment 2:

How did the authors obtain the numbers of the raw materials in page 4, including the strength and loading parameters.

Response:

For the numbers of the raw materials and the strength parameters, from line 168 to 175, we added related explanation and Table 2 to show the beneficial properties of MSC manufactured at the dry mass ratio of 2:1:2 for OSA/FA/SC.

For the loading parameters, from lines 208 to 209,they were decided by the actual stress condition of subgrade construction, and were used by some related studies[1, 2]

Table 2. The physical properties of modified SC (MSC).

Physical properties

OSA: FA: SC

4:3:3

3:4:3

7:5:8

2:1:2

9:3:8

Plastic Limit [%]

27.96

26.94

27.54

20.20

25.77

Liquid Limit [%]

40.51

40.17

42.70

32.6

42.60

Plasticity [%]

12.55

13.23

15.16

12.4

16.83

CBR Value soaking for 96h [%]

17.00

13.00

31.00

40.00

33.00

Comment 3:

How did the authors define plastic limit, liquid limit and plasticity and the optimum moisture content? Please do more explanation.

Response:

Thanks for your suggestions, but we haven’t added related explanation to plastic limit, liquid limit and plasticity and the optimum moisture content, because authors think that they are the general concepts in civil engineering.

So, please you forgive us not to revise them by your suggestions, but if you insist what we should do, please tell us and we will do.

Comment 4:

Its content of Na2O is 2.31%...6.68%, please provide the support materials.

Response:

From lines 110 to 112, we added the testing method and testing department for the mineral compositions and chemical compositions of raw materials.

Comment 5:

Figure 1-3 I cannot see the SEM images detailed information from the pictures including the scales, voltage, etc. I cannot make any comparisons without the size or scale in these images How did the author get the conclusions “

Response:

According to your suggestions, we have added clearer picture to this manuscript for facilitating comparison. It should be noted that, for making picture clearer, we deleted the picture of 20k ×, because the picture of 5k × and 50k × can prove the content from lines 154 to 164.

Comment 6:

The micro structure analysis shows that the OSA and FA possess adsorption capacity and strength. “please add relevant references or provide any experiments result”.

Response:

The micro-structure analysis shows that the OSA and FA possess smaller particle sizes and more pores. Researches [3, 4] show that, for pore medium, the smaller particle sizes and more pores are, the greater specific surface area and adsorption capacity are. So, the OSA and FA have greater specific surface area and adsorption capacity than SC; therefore, they might improve the physical properties of SC like adsorption capacity and strength.

We added the related descript from lines 161 to 164.

Comment 7:

What is the GDS mean? Please add more information about your testing machine

Response:

At line 193 we explain the GDS and its place of production as follows:

The GDS dynamic triaxial testing system, made by GDS (Global Digital System) Instruments Company in England, is used in the study.

Comment 8:

Conclusion section is a bit confusing. The authors should provide better explanation for this

Response:

According to your and others’ suggestions, we revised this manuscript as follows:

In this study, the deformation characteristic of MSC after F-T cycles were measured. The effects of dynamic stress ratio, confining pressure, loading frequency and F-T cycles on the cumulative plastic strain of MSC were investigated. Research results lead to the following conclusions: (1) As the number of cycle loads increases, the axial strain of MSC begins to increase rapidly, then slowly, and finally tends to be stable. The stress-strain curve of MSC conforms to the shake-down theory of materials, in the whole test of cycle loads, testing samples of MSC are in the shake-down stage. The cumulative plastic strain of MSC after 100 cycle loads occupies for 70.87%~78.39% of that after 8000 cycle loads, indicating that MSC possesses remarkable plastic stability after 100 cycles of cycle loads.

 (2) Dynamic stress ratio and F-T cycle are important factors affecting the cumulative plastic strain of MSC after repeated cycle loads. The larger the stress ratio, the larger the axial strain value of MSC under the same number of cycle loads. Under the same low stress ratio, increasing confining pressure and loading frequency have insignificant effect on the cumulative plastic strain of MSC after 8000 loads. The F-T cycles can increase the deformation sensitivity of MSC. Moreover, before and after F-T cycles, the rulesof dynamic stress ratio and confining pressure operate on MSCare same.

(3) For the logarithmic prediction model, when the dynamic stress ratio is high (in this study, it is 1.5 and 2.5), there is a linear relationship between the cumulative plastic strain and piecewise function of load number, and the proposed prediction model has a high correlation coefficient with testing data.

(4) The study shows that the cumulative plastic strain of MSC ranges from 0.38 cm to 2.71 cm. Compared with the related standards [35, 36], the cumulative plastic strain of MSC is well within them, indicating that MSC is a kind of excellent materials used subgrade fillings.

In summary, the deformation characteristic of MSC after F-T cycles indicates that MSC possesses remarkable stability and accords with the related standards of subgrade construction. Therefore, utilizing MSC as subgrade fillings can achieve the purposes of disposing industry solid wastes, thus improving the economic and environmental sustainability of subgrade.

Reference

Rosone, M.; Farulla, C. A.; Ferrari, A., Shear strength of a compacted scaly clay in variable saturation conditions. Acta Geotechnica 2016, 11, (1), 37-50. Yuanqiang Cai; Qi Sun; Lin Guo; C. Hsein Juang; Wang, J., Permanent deformation characteristics of saturated sand under cyclic loading. Canadian Geotechnical Journal 2014, 52, (6), 150214143309001. Kuila, U.; Prasad, M., Specific surface area and pore-size distribution in clays and shales. Geophysical Prospecting 2013, 61, (2), 341-362. Rhee, I.; Kim, Y. A.; Shin, G. O.; Ji, H. K.; Muramatsu, H., Compressive strength sensitivity of cement mortar using rice husk-derived graphene with a high specific surface area. Construction & Building Materials 2015, 96, 189-197.

Reviewer 2 Report

Observations:

Review the general wording of the text, there are some words that should start with a capital letter. The graphs, especially figures 5 and 6, are not appreciated due to the amount of data, it is recommended to look for the way they can be observed better. The research is interesting, but it is suggested that you support more references and research. Consider that more information should be contributed on how research contributes to sustainability, as a complement to technical data.

Author Response

Responses to Reviewers’ Comments on Manuscript ID sustainability-518690

Dear Reviewer 2,

We would like to express our sincere gratitude your thoughtful comments and helpful suggestions for improving the quality of this paper. We have revised the manuscript entitled " Experimental Research on Deformation Characteristics of Using Silty Clay Modified by Oil Shale Ash and Fly Ash as the Subgrade Material after Freeze-Thaw Cycles " (sustainability-573955) according to your comments. Below are point-by-point responses to your comments. The corresponding modifications and corrections were made and highlighted in red in the revised manuscript (MS).

If there is any question regarding this version of the manuscript, please let us know. We are looking forward to receiving your evaluation.

Please you download the PDF

Best regards,

Fuyu Wang and Yangpeng Zhang

Jilin University

Changchun, China

Comment 1:

Review the general wording of the text, there are some words that should start with a capital letter.

Response:

We have revised a large number of words in this manuscript.

Comment 2:

The graphs, especially figures 5 and 6, are not appreciated due to the amount of data, it is recommended to look for the way they can be observed better.

Response:

Thanks very much for your suggestions. However, we didn’t revise the Figures 5 and 6, because it is important to utilize them to show that deformation of MSC conforms to the plastic shake down theory, and to explain the change rules of rebound strain, total axial strain and cumulative plastic strain under cycle loads.

So, please you forgive us not to revise Figures 5 and 6 by your suggestions, but if you insist what we should do, please tell us and we will do.

Comment 3:

The research is interesting, but it is suggested that you support more references and research.

Response:

According to your suggestion, we have added some references in Introduction and the context to support our research.

Comment 4:

Consider that more information should be contributed on how research contributes to sustainability, as a complement to technical data.

Response:

From lines 535 to 538, in the last part of the Conclusion, we added “summary” to show how research contributes to sustainability as follows:

In summary, the deformation characteristic of MSC after F-T cycles indicates that MSC possesses remarkable stability and accords with the related standards of subgrade construction. Therefore, utilizing MSC as subgrade fillings can achieve the purposes of disposing industry solid wastes, thus improving the economic and environmental sustainability of subgrade.

Reviewer 3 Report

Review Report

This experimental work examined the mechanical properties of a mixed material as the subgrade materials using silty clay, oil shale ash and fly ash. They identified a series of important factors and conducted detailed experiments to test the property changes as a function of these factors. I would say it is a fairly well-designed study and they reached a good amount of positive outcomes. It is suitable to be published in this journal in their outlined scope.

Scientifically, the data are rich and the elaboration is Okay. However, on the level of English, some languages and grammar are weak. Another round of language editing is highly recommended after they address my suggestions.

I have the following comments for authors to reconsider the manuscript:

Overall evaluation

The authors used too many "which" and ";" in between sentences. Most of the time, the sentences can be finalized by the period ".", the ";" can be replaced by "and". Because there are too many "which" and ";", sentences are extremely long. This hampers the readability of the paper.

Specific comments - Grammar and narrative

Line 12: dispose -> disposing

Line 24: have little effect -> have insignificant effect

Line 27: remove the "and"

Line 27: prediction -> predicted

Line 28: prediction -> predicted

Line 28-29: are less than the allowable values -> are within the requirements

Line 30: the subgrade materials.

Line 46: good effect -> positive effect

Line 53: would -> can

Line 54: Abdulla et al. [9], however, this is not matching the [9] in References.

Line 57: researches -> studies

Line 61: encouraging -> promising

Line 68: no studies or projects can be referred to estimate -> no studies or projects have estimated

Line 75-76: causing bad effects -> otherwise causing negative effects

Line 77: remove bad

Line 81: remove is also

Line 82: remove based on....,

Line 82: mastering -> understanding

Line 85-94: since you are aiming, you only have to list the hypotheses you will be testing or the goals in this study, instead of a detailed what you have done.

Line 94: the subgrade materials.

Line 98: remove abundant

Line 101: the -> a

Line 106-107: and consists of a large number of inorganic compounds -> dominated by inorganic compounds

Line 112: various compounds, like SO3 -> other compounds including

Why OSA does not have the loss on ignition and a grain density like FA?

Line 114: figure -> figures

Line 115: remove technology

Line 124: structure -> structures

Line 126: different sizes: but how different? How large? What size?

Figure 1-3: without scale bars, it is hard to tell the particle sizes.

Line 129: fine pore: this is too general. The pore are in different sizes to me.

Line 131: remove outline is clear

Line 136: remove foundation

Line 138: The abbreviation of CBR is not clear. It is not defined anywhere.

Line 138: serious -> series?

Line 140: Firstly -> First

Line 143: remove after mixing OSA, FA and SC

Line 144: remove to

Line 145: The preparation of samples -> The sample preparation

Line 145: literature -> Ref

Line 146: remove and

Line 147: research -> study

Section 2.2: can you add some description or a picture of what the samples look like? You have a mixed powder then water is added, so finally is it a brick-like or mud-like? Any post-treatment for it, such as air dry?

Line 152: remove which is

Line 152: adopted -> used

Line 154: remove finishing

Line 156: can response the stress design from GDS: this is unclear.

Line 157: remove to

Line 160: Research -> A previous study

Line 165: number -> numbers

Line 183: load; according -> load. According

Line 187: little influence -> low impact

Line 188: analysis -> analyze

Line 192: should be -> is

Line 193: the total axial strain of testing samples reaches 5% of their height, do you mean: the total axial strain of testing samples leads to 5% change of their height?

Line 194: can't -> cannot

Line 195: response -> output?

Line 195: stress amplitude can’t reach standard, this is hard to understand.

Line 195: and -> or

Use ";" between (1), (2), and (3).

Line 199: remove the two "curve"

Line 202: as the analysis object -> as the representative

Line 203: remove analyzed, as

Figure 5d: Y-axis label is not consistent.

Line 220: in this manuscript -> in this work

Line 220-221: we don’t explain it in detail because the diagram and expressing form of MSC, undergoing 1 and 5 F-T cycles, are similar to those of MSC going through 0 F-T cycles.

Comment: since you didn't show this figure or result, it does not make sense that "we don't explain...". Or you can show that results instead.

Is it fair to say "In addition, the results show MSC undergoing 1 and 5 F-T cycles are similar to those of MSC going through 0 F-T cycle in this work. "

Line 256: number -> numbers

Line 256: in the Sample 1# -> in Sample #1

Line 262: literatures -> studies

Line 263: remove research of

Line 263: take -> takes

Line 264: criteria -> criterion

Line 264-265: the deformation characteristics indicate that MSC possesses better stability performance.

Line 267: big -> large

Line 275-276: testing samples of MSC shows -> testing samples show

Line 277: some impact -> some extent of impact

Line 278: little impact -> insignificant impact

Line 285: and in Table 2, -> In Table 2, (please identify other places you have the similar issues).

Line 286: analysis -> analyses

Line 298: as the research object -> as the goal

Figure 8: why the 1 F-T and 5 F-T cycles show no data at ratio of 0.2? Please insert your explanation.

Line 305: still satisfied? Do you want to say "is still valid"?

Figure 9: you have not defined the k1 or k2 in the figure.

Line 330: Based on the basic condition -> Based on the conditions

Line 343: which is very important to predict -> used to predict

Line 344: literature -> studies

Line 351: adopted -> used

Line 372: Barksdale, R D [31] -> Barksdale [31]

Line 373-376: remove the ";" and so many "and". The sentence is so long.

Line 380-382: the Figure 13, which shows the relation between the cumulative plastic strain of MSC and the cycle load number of log function (base 10), is drawn.

No need to tell the base is 10.

Suggestion: Figure 13 plots the relationship between the cumulative plastic strain of MSC and the cycle load number.

Line 383: I don't think you are applying the log function of cycle load numbers. It only looks like you are using the log scale. No need to mention the "log function".

Line 383: small -> low

Line 386-387: is at the position of log100 -> is corresponding to the cycle load number of 100.

Line 393: converted as follows -> converted to

Line 395: manuscript -> work

Line 396: remove which is

Line 396-397: remove and obtained by Origin 2019 (A trial version),

Line 399: the prediction model adopted are very suitable for predicting -> the model is suitable for predicting

Line 403: remove at

Line 403: less -> smaller

Line 407: Figure 13. The cumulative plastic strain of MSC changes with the cycle load number.

Line 413: Eqn (8) should be (7)

Line 414: Where -> where

Line 416: literature -> studies

Line 418: adopted -> used

Line 428: research -> studies

Line 428-431: almost all the cycle load numbers, used to study permanent deformation characteristics or cumulative plastic strain of soil medium, are lower than one million times. In this manuscript, in order to present the residual deformation in more rigorous cases than previous studies,

Line 432: remove research

Line 433: little -> insignificant

Line 434: lead the worse deformation -> leads to the deformation

Line 435: remove much

Line 436-437: remove to research that deformation of MSC

Line 437: ; furthermore, -> . Furthermore, (please identify similar issues and correct them)

Line 437: prediction -> predicted

Line 438: chose -> chosen

Line 440: ratio -> ratios

Line 440: prediction -> predicted

Line 441: ranges -> range

Line 441: prediction -> predicted

Line 442: are less than the allowable values in the -> are well within the standard of ...

Line 444: remove very

Line 444: material -> materials

Line 450: The specific prediction schemes and results of MSC deformation -> The specific schemes and predicted results of MSC deformation

Table 4: Prediction Results -> Predicted results

Conclusions: please simplify and try to be concise about your conclusions, instead of repeating the texts once again. Instead of summarizing the results, you are only describing what has been done. This is way too tedious. The conclusion has too much repetitive to the main texts.

To me, the 1st paragraph of Conclusions is unnecessary. The (4) is highly repetitive and redundant here. Extract the main results from (1), (2), and (3), instead of pouring repeated texts.

Additional questions:

About the OSA, FA, and SC: were you able to identify the mineralogy of these materials? Or at least the major mineral phases. So that it can be connected to the material's morphology shown in Figures 1-3, in addition to your listed oxide compositions.

References:

#1: if this is a book, then the publishing house and place are missing.

#14: page numbers are missing.

#15: page numbers are missing.

#25: page numbers are missing.

#33: page numbers are missing.

Author Response

Responses to Reviewers’ Comments on Manuscript ID sustainability-518690

Dear Reviewer 3,

We would like to express our sincere gratitude your thoughtful comments and helpful suggestions for improving the quality of this paper. We have revised the manuscript entitled " Experimental Research on Deformation Characteristics of Using Silty Clay Modified by Oil Shale Ash and Fly Ash as the Subgrade Material after Freeze-Thaw Cycles " (sustainability-573955) according to your comments. Below are point-by-point responses to your comments. The corresponding modifications and corrections were made and highlighted in red in the revised manuscript (MS).

If there is any question regarding this version of the manuscript, please let us know. We are looking forward to receiving your evaluation.

Please you download the PDF

Best regards,

Fuyu Wang and Yangpeng Zhang

Jilin University

Changchun, China

Comment 1:

The authors used too many "which" and ";" in between sentences. Most of the time, the sentences can be finalized by the period ".", the ";" can be replaced by "and". Because there are too many "which" and ";", sentences are extremely long. This hampers the readability of the paper.

Response:

We have revised this manuscript according to your suggestions

Comment 2:

Specific comments for Grammar

Response:

We have revised this manuscript by your suggestions about Grammar. Thank you very much for helping us find grammatical errors in this manuscript.

Comment 3:

Abdulla et al. [9], however, this is not matching the [9] in References.

Response:

We have corrected the mistake by your suggestion.

Comment 4:

Line 85-94: since you are aiming, you only have to list the hypotheses you will be testing or the goals in this study, instead of a detailed what you have done.

Response:

From line 93 to 105, we have revised this manuscript as follows:

For better understanding the deformation characteristics of using silty clay modified by OSA and FA as the subgrade material in seasonally frozen regions and thus provide valuable guidance for its practical application, this paper aims to  (1) present the change characteristics on the total axial strain and cumulative plastic strain of modified silty clay (MSC); (2) discuss the effects of dynamic stress ratio, confining pressure, loading frequency and F-T cycles on the cumulative plastic strain of MSC, and corresponding to the above condition, present the cumulative plastic strain equation; (3) obtain the prediction values on the cumulative plastic strain of MSC, and compared with corresponding standard to confirm the excellent performance of MSC used as the subgrade materials.

Comment 5:

Line 126: different sizes: but how different? How large? What size? Figure 1-3: without scale bars, it is hard to tell the particle sizes.

Response:

From line 144 to 152, According to your suggestions, we have added clearer picture to this manuscript for facilitating comparison. It should be noted that, for making picture clearer, we deleted the picture of 20k ×, because the picture of 5k × and 50k × can prove the content from lines 154 to 164.

Comment 6:

Line 129: fine pore: this is too general. The pore are in different sizes to me.

Response:

   At line 158, we have removed “fine” to avoid confusion.

Comment 7:

Section 2.2: can you add some description or a picture of what the samples look like? You have a mixed powder then water is added, so finally is it a brick-like or mud-like? Any post-treatment for it, such as air dry?

Response:

Line 177 to 183, we have added related descriptions to explain what the samples look like and its post-treatment as follows:

First, the OSA, FA and SC are dried in a drying oven at 105–110 ℃ for 24 h and then were cooled in a desiccator to room temperature. Second, the OSA, FA and SC were mixed in a mass ratio of 2:1:2, and pure water was added with the moisture content of 12.95% by several times to obtain mixed samples with the target moisture content. This mixing process is achieved by mixing manually, and after mixing process, the mixed samples must be powder-like. Finally, the mixed samples were placed in a humidor to let moisture to diffuse into the samples evenly.

Comment 6:

Line 156: can response the stress design from GDS: this is unclear.

Response:

At line 198, we revised “can response the stress design from GDS” To “can reach the stress design from GDS”.

Comment 7:

Line 193: the total axial strain of testing samples reaches 5% of their height, do you mean: the total axial strain of testing samples leads to 5% change of their height?

Response:

Line 232, : we revised “reaches 5% of their height” to ” leads to 5% change of their height”

Comment 8:

Line 195: stress amplitude can’t reach standard, this is hard to understand.

Response:

Line 233 to 236, we revised this manuscript as follows:

(3) the experiment apparatus cannot output the setting value of loading stress or stress amplitude. (4) the obvious damage occurs in deformation zone of samples.

Comment 9:

Line 220-221: we don’t explain it in detail because the diagram and expressing form of MSC, undergoing 1 and 5 F-T cycles, are similar to those of MSC going through 0 F-T cycles.

Comment: since you didn't show this figure or result, it does not make sense that "we don't explain...". Or you can show that results instead.

Is it fair to say "In addition, the results show MSC undergoing 1 and 5 F-T cycles are similar to those of MSC going through 0 F-T cycle in this work. "

Response:

From lines 260 to 261, according to your suggestions, we used "In addition, the results show MSC undergoing 1 and 5 F-T cycles are similar to those of MSC going through 0 F-T cycle in this work."

Comment 10:

Figure 8: why the 1 F-T and 5 F-T cycles show no data at ratio of 0.2? Please insert your explanation.

Response:

It is because, for the 1 F-T and 5 F-T cycles, the experiment at dynamic stress of 0.2 don’t include in the experimental scheme listed in Table 2. In fact, it is obvious that, without data at ratio of 0.2, the relation between the ultimate cumulative plastic strain and dynamic stress ratio of MSC in Figure 8 still be easy to be observed.

Comment 11:

Figure 9: you have not defined the k1 or k2 in the figure.

Response:

Line 371: We have deleted k1 and k2 in figure because, in fact, we don’t explain the experimental results by k1 and k2 in the context.

Comment 12:

Suggestion: Figure 13 plots the relationship between the cumulative plastic strain of MSC and the cycle load number.

Line 383: I don't think you are applying the log function of cycle load numbers. It only looks like you are using the log scale. No need to mention the "log function".

Line 386-387: is at the position of log100 -> is corresponding to the cycle load number of 100.

Response:

From lines 427 to 435, we have revised this manuscript by your suggestion.

Comment 13:

Figure 13. The cumulative plastic strain of MSC changes with the cycle load number.

Response:

Line 455, we have revised this manuscript by your suggestion.

Comment 14:

Conclusions: please simplify and try to be concise about your conclusions, instead of repeating the texts once again. Instead of summarizing the results, you are only describing what has been done. This is way too tedious. The conclusion has too much repetitive to the main texts.

To me, the 1st paragraph of Conclusions is unnecessary. The (4) is highly repetitive and redundant here. Extract the main results from (1), (2), and (3), instead of pouring repeated texts.

Response:

According to your and others’ suggestions, we revised this manuscript as follows:

In this study, the deformation characteristic of MSC after F-T cycles were measured. The effects of dynamic stress ratio, confining pressure, loading frequency and F-T cycles on the cumulative plastic strain of MSC were investigated. Research results lead to the following conclusions: (1) As the number of cycle loads increases, the axial strain of MSC begins to increase rapidly, then slowly, and finally tends to be stable. The stress-strain curve of MSC conforms to the shake-down theory of materials, in the whole test of cycle loads, testing samples of MSC are in the shake-down stage. The cumulative plastic strain of MSC after 100 cycle loads occupies for 70.87%~78.39% of that after 8000 cycle loads, indicating that MSC possesses remarkable plastic stability after 100 cycles of cycle loads.

 (2) Dynamic stress ratio and F-T cycle are important factors affecting the cumulative plastic strain of MSC after repeated cycle loads. The larger the stress ratio, the larger the axial strain value of MSC under the same number of cycle loads. Under the same low stress ratio, increasing confining pressure and loading frequency have insignificant effect on the cumulative plastic strain of MSC after 8000 loads. The F-T cycles can increase the deformation sensitivity of MSC. Moreover, before and after F-T cycles, the rulesof dynamic stress ratio and confining pressure operate on MSCare same.

(3) For the logarithmic prediction model, when the dynamic stress ratio is high (in this study, it is 1.5 and 2.5), there is a linear relationship between the cumulative plastic strain and piecewise function  of load number, and the proposed prediction model has a high correlation coefficient with testing data.

(4) The study shows that the cumulative plastic strain of MSC ranges from 0.38 cm to 2.71 cm. Compared with the related standards [35, 36], the cumulative plastic strain of MSC is well within them, indicating that MSC is a kind of excellent materials used subgrade fillings.

In summary, the deformation characteristic of MSC after F-T cycles indicates that MSC possesses remarkable stability and accords with the related standards of subgrade construction. Therefore, utilizing MSC as subgrade fillings can achieve the purposes of disposing industry solid wastes, thus improving the economic and environmental sustainability of subgrade.

Comment 15:

Additional questions:

About the OSA, FA, and SC: were you able to identify the mineralogy of these materials? Or at least the major mineral phases. So that it can be connected to the material's morphology shown in Figures 1-3, in addition to your listed oxide compositions.

Response:

Line 133, we have added the Table 1 to show the mineral compositions of raw materials as follows:

Table 1. The mineral compositions of raw materials

Samples

The mineral compositions

Quartz

Anorthose

Potassium feldspar

Analcime

Calcitum

SC

50%

12%

2%

/

/

FA

30%

/

/

/

/

OSA

25%

8%

/

5%

10%

Kaolinite

Illite/montmorillonite

Mullite

Organic matter

Non-crystalline

SC

5%

30%

/

1%

/

FA

/

/

10%

/

60%

OSA

/

48%

/

4%

/

Notes: (1) OSA = oil shale ash; (2) FA = fly ash; (3) SC = silty clay; (4) “/” means that the mineral content is too small to be detected.

Comment 16:

References:

#1: if this is a book, then the publishing house and place are missing

Line 763, we have corrected this error.

#14: page numbers are missing.

#15: page numbers are missing.

The Ref 14 and 15 become to 22 and 23 after revised manuscript. We have added page numbers to them and it is should note that the page numbers of MDPI publisher is a number.

#25: page numbers are missing.

#33: page numbers are missing.

The Ref 25 and 33 become to 37 and 45, respectively. At lines 638 and 667, we have corrected those error.

Round 2

Reviewer 1 Report

no comments 

Author Response

Responses to Reviewers’ Comments on Manuscript ID sustainability-518690

Dear Reviewer 1,

We would like to express our sincere gratitude your thoughtful comments and helpful suggestions for improving the quality of this paper. We have revised the manuscript entitled " Experimental Research on Deformation Characteristics of Using Silty Clay Modified by Oil Shale Ash and Fly Ash as the Subgrade Material after Freeze-Thaw Cycles " (sustainability-573955) according to your comments. Below are point-by-point responses to your comments.

If there is any question regarding this version of the manuscript, please let us know. We are looking forward to receiving your evaluation.

Please you download the PDF

Best regards,

Fuyu Wang and Yangpeng Zhang

Jilin University

Changchun, China

Comment 1:

Does the introduction provide sufficient background and include all relevant references?

Must be improved

Response:

Thanks for your suggestions, but we haven’t revised Introduction because we think it is suitable for this article.

So, please you forgive us not to revise them by your suggestions, but if you insist what we should do, please tell us and we will do.

Reviewer 3 Report

Review Report - 2nd round

Line 64: known universally -> well-known

Line 64: combustion by product of coal -> byproduct of coal combustion

Line 65: environment -> environment as nanoparticles

Line 66: commonly stabilization -> common

Line 69: CBR appears for the first time, but was not defined.

Line 74-75: California bearing ratio -> can be replaced by "CBR" if you define it earlier.

Line 101: and corresponding to the above condition, present the cumulative plastic strain equation -> present the cumulative plastic strain equation corresponding to the above conditions;

Line 102: prediction -> predicted

Line 102-103: compared with corresponding standard -> compare with national standards

Line 135: get -> obtain

Figures 1-3: (b) labeling is missing.

Line 200-201: show that smaller particle sizes and higher porosity lead to greater specific surface area and adsorption capacity.

Line 349-350: . Results indicate that ..... In addition,

Line 225-227: Table 2 lists the physical properties of MSC obtained from Refs [XX, XX, XX]. The dry mass ratio of 2:1:2 for OSA/FA/SC and moisture content of 12.95% used in this study can be referred to Refs [XX, XX, XX].

Table 2: You should insert reference numbers in the table accordingly.

Line 694-695: delete "in the whole test of cycle loads, testing samples of MSC are in the shake down stage".

Line 698-699: delete "Dynamic stress ratio and F-T cycle are important factors affecting the cumulative plastic strain of MSC after repeated cycle loads."

Line 703: rules? or outcome?

Line 703: are the same.

Line 709: related standards [35, 36] -> national standards [47,48] (I believe the standards now are Refs 47, 48).

Line 709: well under the thresholds,

Line 710: used as

Author Response

Responses to Reviewers’ Comments on Manuscript ID sustainability-518690

Dear Reviewer 3,

We would like to express our sincere gratitude your thoughtful comments and helpful suggestions for improving the quality of this paper. We have revised the manuscript entitled " Experimental Research on Deformation Characteristics of Using Silty Clay Modified by Oil Shale Ash and Fly Ash as the Subgrade Material after Freeze-Thaw Cycles " (sustainability-573955) according to your comments. Below are point-by-point responses to your comments. The corresponding modifications and corrections were made and highlighted in red in the revised manuscript (MS).

If there is any question regarding this version of the manuscript, please let us know. We are looking forward to receiving your evaluation.

Please you download the PDF

Best regards,

Fuyu Wang and Yangpeng Zhang

Jilin University

Changchun, China

Comment 1:

Specific comments for Grammar

Response:

We have revised this manuscript by your suggestions about Grammar. Thank you very much for helping us find grammatical errors in this manuscript.

Comment 2:

Line 69: CBR appears for the first time, but was not defined.

Response:

Line 71: CBR value-> California Bearing Ratio (CBR) value

Comment 3:

Line 74-75: California bearing ratio -> can be replaced by "CBR" if you define it earlier.

Response:

Line 76: We have revised words by your suggestions.

Comment 4:

Figures 1-3: (b) labeling is missing.

Response:

Figures 1-3: We have added this labeling.

Comment 5:

(1) Line 225-227: Table 2 lists the physical properties of MSC obtained from Refs [XX, XX, XX]. The dry mass ratio of 2:1:2 for OSA/FA/SC and moisture content of 12.95% used in this study can be referred to Refs [XX, XX, XX].

(2) Table 2: You should insert reference numbers in the table accordingly.

Response:

(1) Line :225-227:We have revised this manuscript according to your suggestions.

(2) Line 240: The related references have been added into Table 2.

Comment 6:

Line 694-695: delete "in the whole test of cycle loads, testing samples of MSC are in the shake down stage".

Response:

 Line 710: we have deleted them.

Comment 7:

Line 703: rules? or outcome?

Response:

Line 716: Authors replaced “rules” with “outcome”.

Comment 8:

Line 709: related standards [35, 36] -> national standards [47,48] (I believe the standards now are Refs 47, 48).

Response:

Line 723: Thank for your suggestions, we have revised this error by your suggestions.

Comment 9:

Line 709: well under the thresholds,

Response:

Line 723: We have correct words by “well under the thresholds”.
